# Onset of Senescence and Steatosis in Hepatocytes as a Consequence of a Shift in the Diacylglycerol/Ceramide Balance at the Plasma Membrane

**DOI:** 10.3390/cells10061278

**Published:** 2021-05-21

**Authors:** Gergana Deevska, Patrick P. Dotson, Mihail Mitov, D. Allan Butterfield, Mariana Nikolova-Karakashian

**Affiliations:** 1Department of Physiology, University of Kentucky College of Medicine, Lexington, KY 40536, USA; gdeevska@idahocom.org (G.D.); ppdots4@yahoo.com (P.P.D.2nd); 2Markey Cancer Center, Redox Metabolism Shared Resource Facility, University of Kentucky, Lexington, KY 40536, USA; mmitov@idahocom.org (M.M.); david.butterfield@uky.edu (D.A.B.); 3Department of Chemistry, University of Kentucky, Lexington, KY 40506, USA

**Keywords:** sphingomyelin synthase, diacylglycerol, ceramide, mitochondria, lipid droplets, senescence, protein kinase C, AMP kinase

## Abstract

Ceramide and diacylglycerol (DAG) are bioactive lipids and mediate many cellular signaling pathways. Sphingomyelin synthase (SMS) is the single metabolic link between the two, while SMS2 is the only SMS form located at the plasma membrane. SMS2 functions were investigated in HepG2 cell lines stably expressing SMS2. SMS2 overexpression did not alter sphingomyelin (SM), phosphatidylcholine (PC), or ceramide levels. DAG content increased by approx. 40% and led to downregulation of DAG-dependent protein kinase C (PKC). SMS2 overexpression also induced senescence, characterized by positivity for β-galactosidase activity and heterochromatin foci. HepG2-SMS2 cells exhibited protruded mitochondria and suppressed mitochondrial respiration rates. ATP production and the abundance of Complex V were substantially lower in HepG2-SMS2 cells as compared to controls. SMS2 overexpression was associated with inflammasome activation based on increases in IL-1β and nlpr3 mRNA levels. HepG2-SMS2 cells exhibited lipid droplet accumulation, constitutive activation of AMPK based on elevated ^172^Thr phosphorylation, increased AMPK abundance, and insensitivity to insulin suppression of AMPK. Thus, our results show that SMS2 regulates DAG homeostasis and signaling in hepatocytes and also provide proof of principle for the concept that offset in bioactive lipids’ production at the plasma membrane can drive the senescence program in association with steatosis and, seemingly, by cell-autonomous mechanisms.

## 1. Introduction

Ceramide and diacylglycerol (DAG) are the archetypical lipid second messengers that mediate cellular responses to growth factors, cytokines, and other extracellular signals. In a healthy state, the generation and turnover of these two bioactive lipids are strictly regulated to ensure a controlled, transient rise in their local concentrations followed by the activation of downstream signaling molecules such as protein kinase C (PKC) [1], which is regulated by DAG, or protein phosphatase 2A (PP2A) [2] and PKCζ [3], which are among the few identified direct ceramide targets. In principle, DAG is involved in pro-survival, proliferative, and metabolic signals, while ceramide participates in the cellular response to stress and in the onset of cell death.

DAG and ceramide are produced along distinct metabolic pathways. At the plasma membrane, phospholipase C (PLC) generates DAG via the hydrolysis of polyphosphoinositol-4,5-bisphosphate (PIP2), while neutral sphingomyelinase-2 (nSMase-2) catalyzes the turnover of plasma membrane sphingomyelin (SM) to ceramide. The sphingomyelin synthase (SMS) (Phosphatidylcholine:ceramide cholinephosphotransferase, EC 2.7.8.27) is the only metabolic link between ceramide and DAG. SMS transfers the phosphorylcholine group from phosphatidylcholine (PC) to ceramide, producing SM and DAG. SMS is the only enzyme with the capacity to simultaneously affect the homeostasis of these two main bioactive lipids. Three forms of SMS, which differ in their subcellular location, have been identified [4]. SMS1 is responsible for the generation of the bulk of cellular SM and is localized in the cis-Golgi apparatus, while SMS2 is found at the Golgi and at the plasma membrane. A third form, SMSr, exists in the endoplasmic reticulum and catalyzes the synthesis of ceramide phosphoethanolamine, a compound that can be converted to SM by a three-step methylation reaction at the primary amine of the ethanolamine head group.

Sphingomyelin synthase-2 (SMS2, gene name *sgms2*) is a 365-aa integral membrane protein with six transmembrane domains and a large C-terminal cytosolic tail, which is anchored to the plasma membrane via palmitoylation at four cysteine residues [5]. SMS2 seems to localize to the lipid rafts [6]. Studies using SMS2-deficient mice delineated a possible role of the enzyme in tumorigenesis, metabolic disorders, and fatty liver disease. SMS2 deficiency in mice suppresses NF-κB activation in fibroblasts [7] and TLR-4 signaling in microglia [8]. SMS2 deficiency also alleviates the lipid-induced insulin resistance of fibroblasts [9], improves the hepatic insulin response, and suppresses colonic inflammation in response to challenge with dextran sulfate sodium [10]. Metabolic stress has been shown to elevate SMS2 protein abundance [9], while SMS2 overexpression has been found to promote insulin resistance in mice [9] and epithelial–mesenchymal transition in breast cancer cell lines [11]. Mice overexpressing SMS2 specifically in hepatocytes develop spontaneous steatosis, while SMS2-deficient mice exhibit the opposite phenotype [12]. SMS2 overexpression has also been associated with increased expression of inflammatory biomarkers in the aorta, with endothelial dysfunction, and with atherosclerotic plaque instability [12].

The aforementioned studies have indicated that SMS2 activity plays a role in the onset of hepatic steatosis. To begin elucidating the mechanisms behind these effects, we established HepG2 cell lines that constitutively express SMS2 (HepG2-SMS2) or a respective empty control vector (HepG2-EV). Surprisingly, the functional characterization of these cell lines provided evidence that SMS2 overexpression triggered the onset of cellular senescence, characterized by senescence-associated β-galactosidase (SA-β-gal) positivity, chromatin condensation, and mitochondrial dysfunction. The onset of senescence was also accompanied by spontaneous accumulation of lipid droplets and a shift in the homeostasis of key signaling molecules, such as PKC and AMPK. Together, these data indicate that a shift in bioactive lipid homeostasis at the plasma membrane has a profound effect on cell physiology and interferes with cell-autonomous pathways regulating hepatocyte senescence and steatosis.

## 2. Materials and Methods

### 2.1. Materials

N-Hexanoyl-(N-(7-nitrobenz-2-oxa-1,3-diazol-4-yl) amino)-sphingosine (C6-NBD-Cer, cat. #N1154) and N-Hexanoyl-Sphingosine-1 phosphocholine (C6-NBD Sphingomyelin, cat. #N3524) were purchased from Invitrogen (Carlsbad, CA, USA). Geneticin sulfate powder (G418, potency ≥700 µg/mg, cat. #SC29065B) was obtained from Santa Cruz Biotechnology (Dallas, TX, USA). Essentially fatty acid-free bovine serum albumin (BSA, cat. #A6003), triolein (cat. #T7140), and Oil red-O (cat. #O0625) dye were purchased from Sigma-Aldrich (Saint Louis, MO, USA). High-performance thin-layer chromatography (HP-TLC) silica gel 60 plates (cat. #1133260001) were obtained from Millipore Sigma (Saint Louis, MO, USA). A Lowry total protein determination kit (RC DC protein assay, cat. #5000121) was obtained from Bio-Rad (Hercules, CA, USA). Antibodies for the mitochondrial enzymes were from Molecular Probes (Eugene, OR, USA) as follows: Complex I (39kDa subunit, cat. #A21344, used at a final concentration of 0.5 µg/mL), Complex III (subunit core 1, cat. #A21362, final concentration 1 µg/mL), Complex IV (subunit IV, cat. #A21348, final concentration 0.5 µg/mL), and Complex V (Subunit α, cat. #A21350, final concentration 0.5 µg/mL). The VDAC antibody (NDUFA9, cat. #PA5-67275, final concentration of 0.04 µg/mL), MitoTracker Red (cat. #M22425, dilution 1:2500), and MitoTracker Green (cat. #M7514, dilution 1:500) were obtained from Invitrogen (Carlsbad, CA, USA). Antibodies against phospho-PKCα/β (Thr636/641, cat. #9375, dilution 1:1000), phospho-AMPKα (Thr172, D4D6D, cat. #50081, dilution 1:1000), AMPKα (D5A2, cat. #5831, dilution 1:1000), and phospho-PKD (Ser744/748, cat. #2054, dilution 1:1000) were obtained from Cell signaling (Danvers, MA, USA). All other reagents were from Fisher Scientific (Pittsburgh, PA, USA).

### 2.2. Cloning of Full-Length Human V5-Tagged Sgms2 and Establishing of Stable Cell Lines

For cloning the human sphingomyelin synthase 2 (HGNC:28395, NM 152621), a cDNA clone (MGC:26963, IMAGE:4823252) was purchased from Open Biosystems (cat. #MHS1010-9203808) and PCR amplified using the primers 5′-CCAAACTAGTGCCATGGATATCATAGACAGCAA-3′ (F) and 5′-GATTACGCGTGGTCGATTTCTCATTGTCTTCACC-3′ (R), where the underlined sequences correspond to the 5′ and the 3′ end of sgms2 cDNA. The ORF sequence encoding full-length human sgms2 (minus the stop codon to allow for V5 fusion at the C-terminus) was cloned into a pcDNA3.1/V5-His-TOPO vector containing Neomycin selection marker (Invitrogen, Carlsbad, CA, USA). The resulting sgms2-pcDNA3.1/V5-His-TOPO (SMS2-pcDNA) plasmid was used to transfect HepG2 cells obtained from ATTC (Manassas, VA, USA). Cells maintained in Minimal Essential Medium (MEM, Invitrogen, Carlsbad, CA, USA) supplemented with 10% FBS and 1% penicillin/streptomycin (Invitrogen, Carlsbad, CA, USA) in a humidified atmosphere of 95% air and 5% CO2 at 37 °C were initially transfected with 1 μg/1.0 × 10^6^ of SMS2-pcDNA or empty vector control plasmid (EV-pcDNA) using FuGENE^®^ HD transfection reagent (Promega, Madison, WI) following the manufacturer’s instructions, resulting in approximately 50% transfected cells in 48 h. Stable clones were selected in the growth medium containing 2 mg/mL geneticin (G418, specific activity > 700 µg/mg) under continuous pressure for 3 weeks. Single-cell colonies were established and expanded in the presence of G418 (2 mg/mL). Expression of the SMS2 protein was examined by indirect immunofluorescence staining.

A colony derived from a single cell with the expected subcellular localization and the highest expression of SMS2 protein was used throughout the experiments. After initial propagation, aliquots containing 250,000 cells on average were frozen and referred to as passage “0”. Cells were maintained for no more than 16–20 passages. To ensure that the observations were not clonal artifacts, the results shown in Figure 1 were confirmed in HepG2 cells transiently expressing SMS2. The β-galactosidase positivity, inflammasome activation, and lipid droplets accumulation were confirmed in a second single cell-derived clone.

### 2.3. Indirect Immunofluorescence and Cell Staining

Cells were grown on cover slips to sub-confluence (defined as 85–90% confluency) and fixed with 3.7% paraformaldehyde in PBS. After quenching the autofluorescence with 50 mm NH4Cl in PBS, the cells were permeabilized with 0.2% Triton X-100 and then incubated with blocking buffer (0.5% BSA in PBS) for 1 h at room temperature. Incubation with mouse monoclonal anti-V5-tag antibody (Invitrogen, Carlsbad, CA, USA; clone 2F11F7, cat. #37-7500, dilution 1:200) was performed overnight at 4 °C, followed by incubation with anti-mouse FITC-conjugated secondary antibody (1 h at room temperature). Cells were counterstained with 1 μg/mL of rhodamine-labeled wheat germ agglutinin (Vector Labs, Burlingame, CA, USA) to visualize Golgi. Mounting on slides was performed in DAPI-Vectashield mounting medium (Vector Labs, Burlingame, CA, USA). Staining of mitochondria was performed with the fluorescent dyes MitoTracker green (membrane potential independent) and MitoTracker red (membrane potential dependent) in living cells. After 1 h of incubation at 37 °C and 5% CO_2_, cells were washed 3 times to remove any unincorporated dye, fixed, and examined by confocal microscopy. Cells grown on cover slips were fixed and stained with Oil Red-O to reveal neutral lipid accumulation. β-galactosidase staining was performed in fixed sub-confluent cells grown on cover slips and incubated at 37 °C in the dark with SA-β-gal stain solution, pH 6.0, containing 1 mg/mL of X-gal substrate. DAPI staining of nuclear chromatin was performed in fixed cells grown on cover slips mounted with DAPI-containing mounting medium.

### 2.4. Western Blotting

Cells were harvested by scraping and pelleted by centrifugation at 500g for 4 min at 4 °C. Cells were lysed in 50 μL buffer (1 mM EDTA, 1.0% Triton X-100, 1 mM Na_2_VO_4_, 1 mM NaF, 1:100 (*v*/*v*) protease inhibitor cocktail, and 50 mM Tris-HCl (pH 7.4)) on ice for 30 min and centrifuged at 16,000g for 10 min at 4 °C. Proteins (usually 80–100 μg/lane, based on Lowry assay) were resolved by 10% SDS-PAGE and transferred to Immobilon^®^-P polyvinylidene difluoride membrane (Millipore Corporation, Billerica, MA, USA). Protein expression was analyzed using the indicated antibodies at dilutions described in Section 2.1. Protein–antibody interactions were visualized using ECF substrate and a Storm860 Phospho-imager scanning instrument. Data were analyzed and quantified using ImageQuant5.0 software (Molecular Dynamics, Sunnyvale, CA, USA).

### 2.5. RNA Isolation, Reverse Transcription, and Quantitative PCR

Total RNA was extracted from 1 × 10^6^ cells using TRIZOL^®^ reagent (Invitrogen, Carlsbad, CA, USA). Reverse transcription was performed with random hexamers (Roche, Indianapolis, IN, USA) using 2 μg RNA and Superscript II™ reverse transcriptase (Invitrogen, Carlsbad, CA, USA). Quantitative RT-PCR was performed with the following primers: nlrp3-(F): 5′-ggagagacctttatgagaaagcaa-3′, nlpr3-(R) 5′-gctgtcttcctggcatatcaca -3′, IL1β-(F): 5′-ctgtcctgcgtgttgaaaga, IL-1β-(R) 5′ttgggtaatttttgggatctaca-3′, β-actin-(F) 5′-tatggagaagatttggcacc-3′ and β-actin-(R) 5′-gtccagacgcaggatggcat-3′, using JumpStart Taq DNA polymerase (Sigma, Saint Louis, MO, USA). PCR products were separated by electrophoresis on 1.8% agarose gel containing 0.02% ethidium bromide and visualized on a Scion Image gel imaging system. Subsequent quantification and analyses were performed using ImageQuant5.0 software (Molecular Dynamics).

### 2.6. Labeling Experiments

Cells were grown to sub-confluence (90%) in 6-well plates and labeled with ^3^H-palmitic acid (50 mCi/mmol specific activity, American Radiochemical Corporation, St. Louis, MO, USA) for 18 h. Palmitic acid was delivered as a complex with BSA (2:1, by mol) at a final concentration of 0.1 mM. In situ labeling with C6-NBD-Cer at a final concentration of 6 μM was performed as described previously [13]. Cells were harvested, protein content was measured by Lowry assay, and lipids were extracted as described.

### 2.7. Mitochondrial Respiration

A mitochondrial respiration assay was performed in cultured cells (1 × 10^5^/well) using an XF96 extracellular flux analyzer (Seahorse Biosciences). Assays were run in a serum-free medium containing 10 mM glucose, 3 mM glutamine, and 1 mM pyruvate. Oligomycin (1.25 μM), FCCP (1.0 μM), and antimycin A (2.0 μM)/rotenone (2.0 μM) were injected at the indicated timepoints to inhibit components of the electron transport chain. The oxygen consumption rate (OCR) was monitored every 20 min. Basal respiration, maximal respiration, coupling efficiency, and ATP production were calculated using Wave software and XF Report Generators.

### 2.8. Lipid Analysis

Phospholipid levels were analyzed in extracts from cells prepared by the method of Bligh and Dyer, modified as described previously [14]. Lipids were separated by high-performance thin layer chromatography (HP-TLC) on silica gel 60 plates using chloroform:methanol:triethylamine:2-propanol:0.25% potassium chloride (30:9:18:25:6, by vol.) as a developing solvent. The regions corresponding to SM, PC, phosphatidylserine (PS), and phosphatidylethanolamine (PE) were sprayed with 50% sulfuric acid, and the plates were baked at 200 °C for 3.5 h. Inorganic phosphorus was quantified according to Kahovcová and Odavić [15].

For the triacylglycerol (TAG), DAG, and cholesterol assay, lipids were extracted with chloroform/methanol (2:1, by vol.). DAG and TAG were separated by HP-TLC on silica gel 60 plates in chloroform:acetone:acetic acid (95.5:4:0.5, by vol.) [16]. The regions migrating with trioleoyl and dioleoyl standards (Avanti Polar Lipids, Alabaster, AL, USA) were scraped off the plates. Lipids were eluted from the silica with chloroform:methanol:water:acetic acid (100:100:5:0.5, by vol.), dried under vacuum, and dissolved in isopropyl alcohol. TAG and DAG were quantified using the Triglyceride-M kit (Wako, Japan) following the manufacturer’s instructions. Total cholesterol (free and esterified) was determined in a separate aliquot of the same extract according to the method of Sperry and Webb [17].

Sphingolipid analysis was conducted by electrospray ionization with tandem mass spectrometry using an ABI 4000 quadrupole-linear ion trap mass spectrometer [18] with internal standards from Avanti Polar Lipids (Alabaster, AL).

### 2.9. Statistical Analyses

Analyses were conducted using three dishes (either 10 or 30 sm^2^) as replicates per experimental point. After quantification, data were calculated as average +/− SD. Differences associated with SMS2 overexpression were assessed using Student’s *t*-test. All graphs present the average, SD, and statistical significance of changes calculated for the three replicates in an individual experiment. Each experiment was repeated at least two times (some three or four times) using cells at different passage numbers and from different frozen stocks of the same clone. Variations in the exact magnitude of difference were observed between cells of different passages and stocks, but all reported differences remained statistically significant in all experiments. The cause of such variability is likely unrelated to SMS2 overexpression but rather to changes during the freezing/thawing of the stock and factors related to cell density and passaging in culture; therefore, combining the results of all experiments together for statistical analyses was deemed unjustified for the purpose of assessing the impact of SMS2 overexpression.

## 3. Results

### 3.1. Characterization of HepG2-SMS2 Cell Line

HepG2 cells were transfected with sgms2-pcDNA3.1/V5-His-TOPO plasmid or the respective empty vector (EV). Single cell-derived colonies were expanded under G418 selective pressure and are referred to as HepG2-SMS2 or HepG2-EV, respectively. Indirect immunofluorescence studies confirmed that SMS2 was expressed at the Golgi apparatus and at the plasma membrane (Figure 1A), as reported earlier [4]. To confirm that the overexpressed enzyme was functionally active, sphingomyelin synthase activity was measured in cell homogenates using NBD-Ceramide as a substrate [19]. The HepG2-SMS2 cells exhibited eight-fold higher activity than HepG2-EV cells (Figure 1B). Labeling of live cells with precursors, either ^3^H-choline (Figure 1C) or fluorescent NBD-Ceramide (Figure 1D), confirmed the elevated flux though the SMS pathway.

The effect of SMS2 overexpression on lipid steady-state levels was assessed by mass spectrometry (Figure 2). These analyses failed to show the anticipated increases in SM and decreases in ceramide in the HepG2-SMS2 cells. No significant changes were seen in the levels of the other product in the SMS reaction, PC, based on TLC analyses (data not shown). The only metabolite of the SMS reaction for which the differences reached statistical significance was DAG (Figure 2D), which increased by almost 500 pmol/mg, or 50%. One should bear in mind that SM and PC are present in cells at much higher levels than ceramide and DAG (i.e., 10–100 fold); therefore, our analyses may have missed subtle changes in the levels of the less abundant lipids, especially as the magnitude of increases seen for DAG was within the margin of error of the measurements of PC and SM. In addition, compensatory increases in the activity of sphingomyelinase(s) that convert SM to ceramide might explain the lack of net ceramide decline in HepG2-SMS2 cells.

### 3.2. SMS2 Overexpression Downregulates PKC Signaling Pathways

Our observations that SMS2 overexpression did not cause a significant change in the levels of PC and SM, the main structural lipids of the plasma membrane, suggest that gross perturbations of the membrane structure are unlikely. SMS2 effects were seemingly limited to DAG and manifested as a chronic DAG accumulation (Figure 2D). To assess the impact of SMS2 overexpression on membrane signaling, we tested the status of protein kinase C (PKC), for which DAG is a key regulator. SMS2 overexpression was associated with significant downregulation of the levels of phosphorylated PKCα/β in response to the prototypical PKC activator, phorbol myristate acetate (PMA) (Figure 2E). In contrast to these two classical, DAG-dependent PKC isoforms, the phosphorylation pattern of PKCµ, which is DAG independent, was not influenced by SMS2 overexpression. Although transient spikes in DAG levels lead to PKC activation, it has long been known that chronic DAG generation following prolonged activation of phospholipase C induces downregulation of PKC levels [20]. Our findings are consistent with such a scenario and show that an elevated flux through the SMS2 reaction perturbs PKC-mediated signaling in hepatocytes.

### 3.3. Onset of Senescence in HepG2-SMS2 Cells

Initial observations indicated that HepG2-SMS2 cells proliferate slower than HepG2-EV cells do. To quantitatively assess this, the cell proliferation rate was monitored with the CellTrace™ Carboxyfluorescein succinimidyl ester (CFSE) reagent for 5 days (Figure 3). The consecutive generation of live cells from a parental population (indicated in dark blue) was seen as discrete peaks in the histogram represented in Figure 3A. Seven successive generations were observed in the control cells within the experimental period, but only three in HepG2-SMS2 cells. With each division, cell-associated fluorescence decreased by half, and at day 5, HepG2-SMS-2 cells had almost three-fold higher CSFE fluorescence than HepG2-EV cells (Figure 3B), lending further support to the conclusion that SMS2 overexpression resulted in suppressed proliferation rates.

To test whether the delayed proliferation may indicate the onset of senescence, an assay for the senescence-associated β-galactose (SA-β-Gal) activity was conducted. HepG2-SMS2 cells showed a strong positive signal for SA-β-Gal activity at pH 6.0 (Figure 3C, upper right panel). Co-staining with DAPI, a nuclear stain, revealed the appearance of heterochromatin foci at the edges of the nuclei (Figure 3C, lower right panel, indicated with white arrows), an independent characteristic of senescent cells that is linked to repression of proliferation-promoting genes [21].

As a third independent validation for the onset of senescence, we measured the expression of two key genes responsible for the senescence-associated secretory phenotype (SASP). SASP is group of proteins that are secreted by senescence cells and facilitate the onset of inflammation in the tissues of aging organisms [22] and/or exacerbate the senescence phenotypes in a cell-autonomous manner. As shown in Figure 3D, the mRNA expression of IL-1β, the major protein of SASP, and nlfr3, the key component of inflammasome, which initiates SASP, was substantially higher in the HepG2-SMS2 cells as compared to controls. Together, these data clearly show that SMS2 overexpression is associated with the onset of cellular senescence in hepatocytes.

### 3.4. SMS2-Induced Senescence Is Characterized by Defects in Mitochondrial Morphogenesis and in ATP Production, Likely Caused by Degradation of ATP Synthase

Senescent cells exhibit distinct changes in mitochondrial functions. Mitochondrial dysfunction can drive the senescence process by elevating reactive oxygen species (ROS) production, resulting in oxidative stress, DNA damage, and inflammasome activation [23]. Consistently, the HepG2-SMS2 cells exhibited elongated and protruded mitochondria, an indication of a defect in normal fission and fusion events that are all required to meet the metabolic demands of the cells and for the removal of dysfunctional entities [24,25] (Figure 4A). These morphological changes were accompanied by a significant functional decline. Analysis of oxygen consumption rates (Figure 4B–E) showed a reduction in the mitochondrial basal and maximal respiration rates (OCR), the coupling efficiency, and ATP production. The maximal respiration rates and ATP production were most severely affected, while non-mitochondrial respiration was unchanged. Western blot analyses of the complexes forming the electron transport chain (ETC) showed a drastic decrease in the levels of the mitochondrial alpha subunit of complex V (the ATP synthase) and, to a lesser degree, a decline in subunit core I of complex I and in cytochrome C oxidase (complex IV) (Figure 5). In contrast, the abundance of complex III was seemingly unaffected. These findings correspond well to the association between the levels of several mitochondrial complexes and mitochondrial dysfunction seen in hippocampal neural stem cells with Alzheimer disease-associated mutation [26]. Noteworthily, ATP synthase has been shown to undergo age-associated degradation in the brain [27], while ATP subunit β expression is reduced in the livers of type 1 and type 2 diabetic mice [28].

### 3.5. Evidence of Excessive Lipid Droplet Accumulation and AMPK Deregulation in HepG2-SMS2 Cells

Microscopic examination of HepG2-SMS2 cells revealed the presence of vesicles resembling lipid droplets, even in a standard tissue culture medium supplemented with 10% serum as the only potential source of fatty acids. Indeed, these vesicles were positive for Oil Red-O stain, which interacts with neutral lipids, including TAG, DAG, and cholesterol esters (Figure 6A). Measurements of the lipid levels showed that TAG levels were higher in HepG2-SMS2 cells as compared to HepG2-EV cells, while the levels of cholesterol were similar (Figure 6B). To begin evaluating the cause for the excessive lipid accumulation, a detergent-free assay of the activity of DGAT, which acylates DAG to TAG and is a rate-limiting step in TAG synthesis, was performed. Interestingly, HepG2-SMS2 exhibited lower DGAT activity than the control cells (Figure 6C), suggesting that the accumulation of lipid droplets seen in HepG2-SMS2 is unlikely to be due to increased DAG availability and likely represents a more complex phenomenon. The elevated AMPK phosphorylation is in line with the known functions of the kinase as an energy sensor activated by a drop in ATP levels, as seen in HepG2-SMS2 cells. However, from a metabolic standpoint, it is difficult to infer how (if at all) the increased AMPK levels and phosphorylation relate to the substantial lipid droplet accumulation in SMS2-overexpressing cells. This apparent disconnect between AMPK activation and its metabolic targets in HepG2-SMS2 cells could reflect a metabolic reprograming and the onset of a less energetic state that are seen in senescence cells [28]. The apparent constitutive activation of AMPK in HepG2-SMS2 cells, however, parallels well the reported role of this kinase in inducing cell cycle arrest, particularly in activating p53 and p38 [29], which, in turn, has been implicated in the induction of NF-kB, SASP, and senescence [30].

## 4. Discussion

This manuscript provides evidence that offset in the homeostasis of bioactive lipids at the plasma membrane can drive senescence in hepatocytes. Initially defined as a decline in proliferative capacity and gradual onset of permanent growth arrest in culture, the concept of cellular senescence has evolved to include changes in metabolism, signaling, morphology, and secretory patterns [29]. Senescent cells are present in almost all organs of aging organisms and contribute to aging and aging-associated diseases. Both in vitro and in vivo, the onset of senescence is characterized by the presence of SA-β-gal activity and heterochromatin foci; by the activation of DNA damage responses, several tumor suppressor genes, and inflammasome; and by the onset of SASP. Oxidative stress, telomere shortening, mitochondrial dysfunction, and defects in autophagy are the chief culprits for driving senescence in vivo, while oxidative, chemical, and osmotic stress; oncogenes; radiation, and various genotoxic drugs all induce senescence acutely in cultured cells. The studies reported herein suggest that the state of bioactive lipid homeostasis at the plasma membrane, the main signaling compartment of cells, is yet another possible causative factor of senescence.

The adult liver consists mostly of quiescent cells that renew very slowly [30]; however, the liver has a powerful regenerative capacity that decreases during normal aging, ostensibly due to senescence [31]. Senescent hepatocytes are present in the liver. The number of senescent hepatocytes increases in healthy aging, as well as in severe chronic liver disease [32,33,34], fatty liver disease [35], non-alcoholic steatohepatitis [36], and alcoholic liver disease [37] and could reach 84% of all hepatocytes in the final stages of cirrhotic liver disease [38,39,40,41,42]. It has been postulated that the increase in the incidence of senescent cells in disease state drives disease progression by limiting the regenerative capacity of hepatocytes. The mechanisms underlying hepatocyte senescence, however, are incompletely understood.

It is well known that aging has a substantial impact on overall lipid metabolism. An inherent metabolic shift takes place during aging, leading to loss of protein and accumulation of fat in various tissues. Unsaturated fatty acids in the membrane phospholipids become oxidized, causing membrane damage and inflammation [43]. Deregulated lipid metabolism has also been implicated in changing the biophysical properties of the membrane (fluidity, lateral diffusion, and asymmetry), affecting signaling processes, the activity of membrane proteins, and the clearance of apoptotic cells [44,45,46]. The effects of aging on minor lipid metabolites, however, are far less understood. The findings reported herein show that a subtle shift in the balance of DAG/ceramide levels at the plasma membrane is sufficient to drive the senescence program. As the levels of the other two metabolites in the reaction, SM and PC, which are the major structural components of the membranes, were unaffected in our model, the senescence program is seemingly initiated in the absence of gross changes in membrane properties and structure. The effects also seem to be specific for SMS2. This is one of the three known SMS enzymes and the only one that has plasma membrane localization. Although a pool of SMS2 also resides in the Golgi, the observation that HepG2 cells overexpressing SMS1 (which localizes exclusively to Golgi) do not exhibit the senescence phenotype [13] indicates that the effects seen in HepG2-SMS2 cells are attributed to SMS2 functions at the level of the plasma membrane rather than the Golgi.

DAG and ceramide production is strictly controlled in healthy cells. While transient spikes in DAG levels are required for PKC activation during signaling, chronic DAG increases can induce PKC degradation and downregulation as a mechanism postulated to protect PKC-dependent pathways against pathological upregulation of DAG production [20]. An abundance of studies have shown that failure to tightly regulate DAG production has a profound impact on cell functions. The tumorigenic process, for example, has been correlated with increased signaling capacity via DAG-dependent pathways, caused by elevated activities of phospahtidylinositol-3-kinase and PLC, the two enzymes controlling the transient spikes in DAG levels during signaling [47]. Sustained DAG production caused by the deletion of DAG kinase, which converts DAG to PA and thereby negatively regulates the basal DAG levels, has been associated with enhanced activation of the IKK–NF-κB pathway and increased apoptosis in NKT cells [48]. In contrast, suppressed DAG production achieved by overexpressing a membrane-targeted constitutively active version of DAG kinase-α (DGKα) causes T cell defects [48]. Thus, these results support the possibility that the chronic elevation of DAG seen in our model might be a likely cause for the observed senescence phenotype, possibly due to PKC downregulation.

Mitochondrial dysfunction is characteristic of aging and senescence; therefore, understanding the mechanisms of mitochondrial dysfunction in HepG2-SMS2 cells might be key for understanding the mechanisms leading to senescence in our model. PKC deregulation might be one of several involved pathways, since some PKC forms, i.e., PKCα and PKCδ, are involved in the regulation of mitochondrial fission/fusion [49,50]. At the same time, changes in mitochondria lipid composition could also affect mitochondrial biogenesis and functions. Our study did not measure the impact of SMS2 overexpression on SM/PC/DAG/ceramide content at the organelle level. Furthermore, SMS2 is not located in the mitochondria. Nevertheless, in view of the active lipid exchange between mitochondria and ER/Golgi apparatus via membrane contact sites and mitochondria-associated membranes [51], changes in mitochondrial lipid content are likely. Interestingly, mitochondrial SM, PC/PE ratio, and cardiolipin levels have been associated with mitochondrial dysfunction in the context of fatty liver disease [52]. Similarly, aging and senescence have been associated with changes in the lipid content of mitochondria and mitochondria-associated membranes and membrane contact sites [53]. Thus, our observations lend further support to the emerging concept of the functional importance of lipid homeostasis in senescence.

Deregulated ceramide homeostasis might also contribute to the senescence phenotype of SMS2-overexpressing cells. In spite of the elevated flux through the SMS reaction, the ceramide level did not change, indicating possible compensatory activation of nSMase-2 that also localizes to the plasma membrane. In hepatocytes, nSMase-2 is constitutively upregulated with aging (due to elevated oxidative stress in the aging liver) [54,55]; it is a tumor suppressor and is silenced in hepatocellular carcinomas and in some leukemias [56,57]. In 1995, ceramide was implicated in the onset of cellular senescence in human diploid fibroblasts [58], an effect later extended to other cell types [59]. Consequent studies have defined an important role for ceramide in carcinogenesis and apoptosis, diabetes and steatosis, autophagy, and DNA repair [60,61,62,63]. Ceramide has been shown to regulate telomerase activity [64] and p21 [65] and to mediate cytokine signaling among others [66]. Abnormalities in hepatic ceramide metabolism have been implicated in the onset of fatty liver disease and also regulate hepatic autophagy and liver regeneration [67,68,69,70].

Juxtaposing ceramide and DAG metabolism as part of one, seemingly cell-autonomous mechanism that connects hepatocyte senescence and steatosis is another informative point of our study. It is already known that hepatocellular senescence and non-alcoholic fatty liver disease (NAFLD), the clinical manifestation of hepatocyte steatosis, are closely connected [71]. There is a significant overlap in the key features of both. For example, palmitic acid, the culprit of NAFLD, is known to induce inflammasome, while senescent cells have been implicated in the progression of liver disease, although not all “markers” of senescence seem to correlate with NAFLD [36]. Systemic factors such as chronic inflammation and SASP have been shown to drive both hepatocyte senescence and NAFLD. A recent study provided the first in vivo evidence that hepatocyte senescence could drive the development and progression of NAFLD by a mechanism involving mitochondrial dysfunction and decreased lipolysis [72]. DAG and ceramide are key players in steatosis; therefore, by linking disturbances in their homeostasis to senescence, our findings outline a potential new common mechanism of senescence and steatosis in hepatocytes.

## Figures and Tables

**Figure 1 cells-10-01278-f001:**
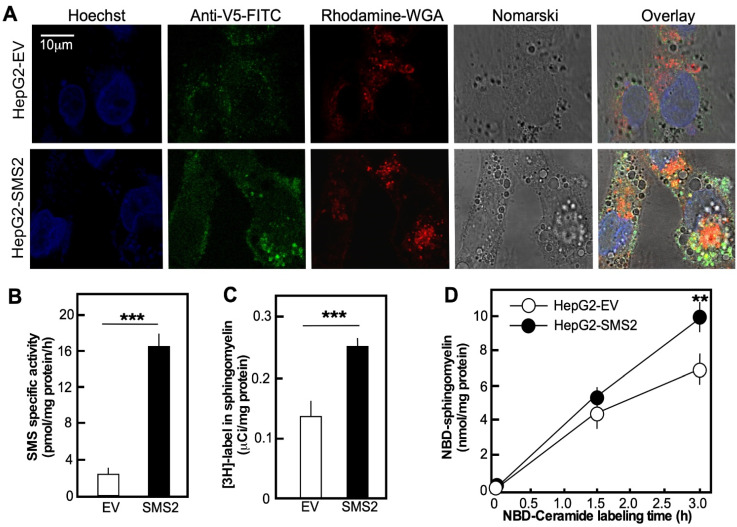
Stable overexpression of functional SMS1 in HepG2 cells. (**A**) Indirect immunofluorescence performed on fixed, permeabilized HepG2-EV and HepG2-SMS2 cells using antibody against the V5 tag of SMS2 protein and WGA conjugated to rhodamine as a marker for Golgi. (**B**) SMS-specific activity measured in whole cell lysates using NBD-Ceramide as a substrate. The values are average+/-SD of three dishes per group. The activity assay for each dish was in quintuplicate, which yielded the same value. (**C**) Incorporation of 3H-palmitate into SM in cells labeled with 0.1mM 3H-palmitate (50 mCi/mmol specific activity) delivered as a complex with BSA (2:1, by mol.) for 18 h. Average ± SD (error bars) are shown (*n* = 3 dishes/group). (**D**) Conversion of NBD-Ceramide to NBD-SM. Cells were labeled with NBD-Ceramide (5 μM) for the indicated time followed by extraction and analyses of NBD-Ceramide and NBD-SM by HPLC. Average ± SD (error bars) are shown (n = 3 dishes/group; ** *p* < 0.05; *** *p* < 0.01). Results are representative for three independent experiments, in which each group was tested in triplicate.

**Figure 2 cells-10-01278-f002:**
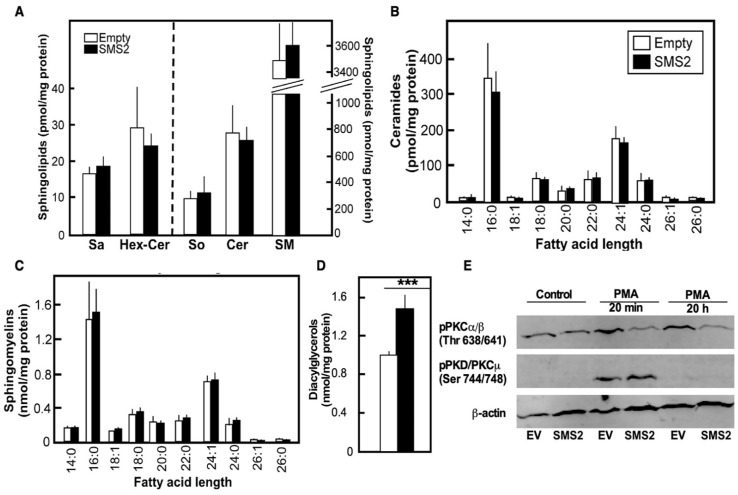
Changes in lipid homeostasis and PKC signaling following SMS1 overexpression. HepG2-EV and HepG2-SMS2 cells were grown to confluency and treated as indicated (**A**–**C**). The levels of the major sphingolipid classes (**A**), ceramides (**B**), and sphingomyelins (**C**) measured in HepG2-SMS2 or HepG2-EV cells by mass spectrometry in one independent experiment. Data are average +/− SD (*n* = 3 dishes). (**D**) Levels of DAG measured using M-type TAG kit (Wako) following lipid extraction and TLC-based separation, as indicated in the “Materials and Methods” section. Data are average +/− SD (*n* = 3 dishes, *** *p* < 0.005). (**E**) Activation patterns of PKC isoforms based on Western blotting-based assessment of phosphorylation. Cells were treated as indicated using three dishes per group. β-actin levels were used as a loading control. The image shown is representative of the Western blotting result observed in two independent experiments, each performed with three dishes per group. Full-sized gel images are provided in the Appendix A.

**Figure 3 cells-10-01278-f003:**
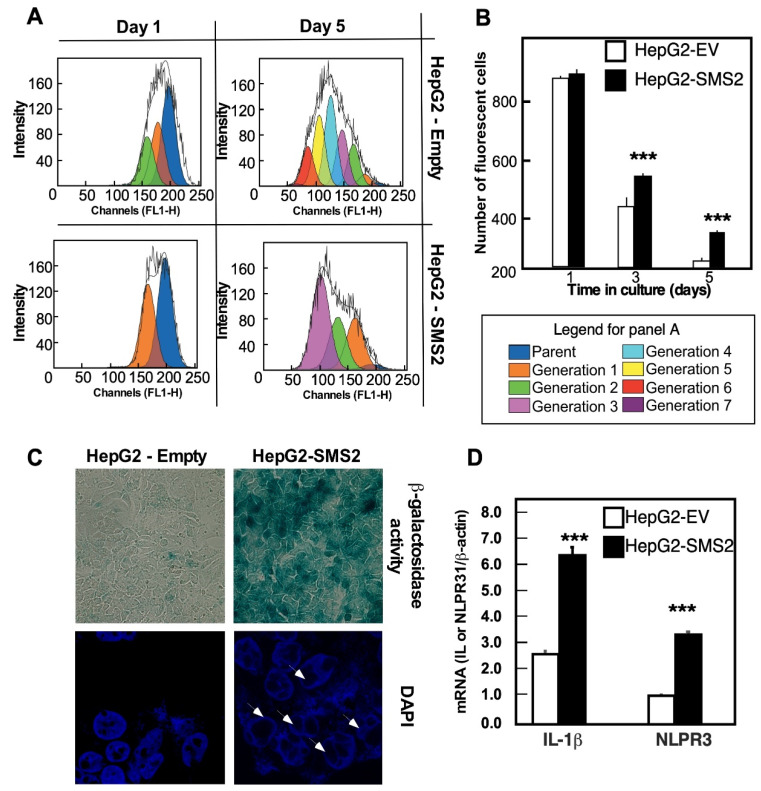
SMS2 overexpression leads to decreased proliferation and senescence. (**A**,**B**) Cell proliferation analyzed with CellTrace™ CFSE Cell Proliferation Kit. Cells were labeled in suspension with CFSE dye for 10 min, plated in 6-well plates, and harvested at the indicated timepoints for flow cytometry analyses. (**C**) SA-β-gal activity and DAPI staining of fixed cells. White arrows indicate heterochromatin foci at the edges of nuclei. (**D**). Expression of inflammasome markers in HepG2-EV and HepG2-SMS2 cells analyzed by RT-PCR. Data are average +/− SD (*n* = 3 dishes, *** *p* < 0.005). Analyses were repeated in two (**A**,**B**,**D**) and three (**C**) independent experiments with similar outcomes.

**Figure 4 cells-10-01278-f004:**
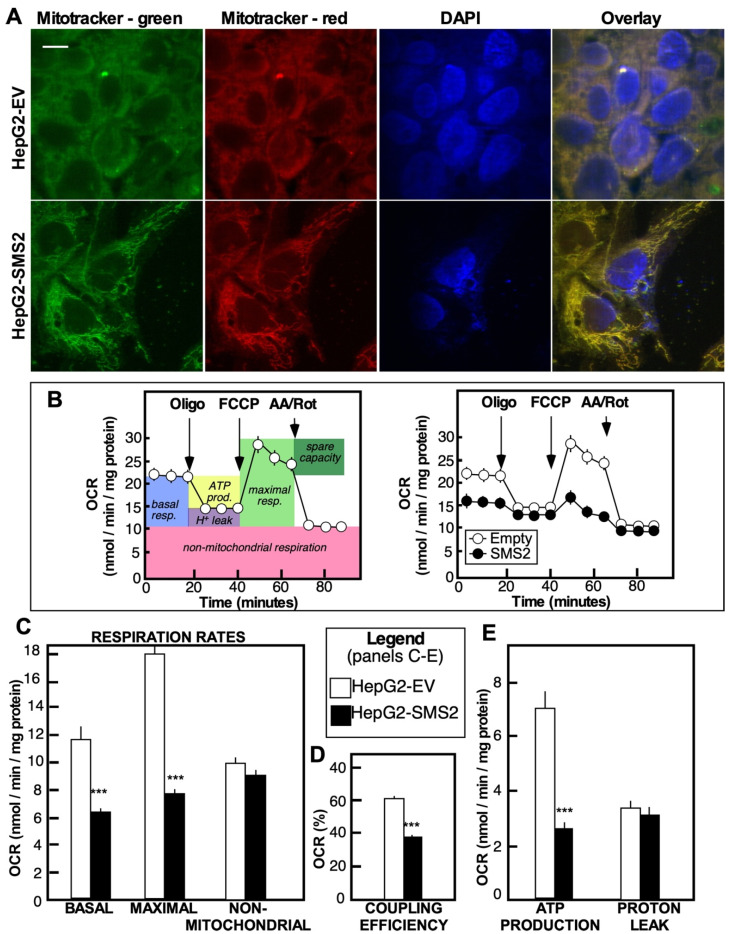
Mitochondrial dysfunction in HepG2-SMS2 cells (**A**) Mitochondria of live HepG2-SMS2 or HepG2-EV cells stained with MitoTracker green (membrane potential independent) and MitoTracker red (membrane potential dependent). Bar = 10 μm. (**B**) Assay of mitochondrial respiration in cultured cells using XF96 extracellular flux analyzer (Seahorse Biosciences) in serum-free medium containing 10 mM glucose, 3 mM glutamine, and 1 mM pyruvate. Inhibitors of the different components of the electron transport chain (1.25 µM oligomycin, 1.0 µM FCCP, and 2.0 µM antimycin A/2.0 µM rotenone) were injected at the indicated times. Oxygen consumption rate (OCR) was measured following the standard assay protocol. (**C**) Basal, maximal, and non-mitochondrial respiration rates. (**D**) Coupling efficiency. (**E**) ATP production and proton leak. Data were calculated using Wave software and XF Report Generators (Seahorse Biosciences) and are the average +/− SD (*n* = 3 dishes, *** *p* < 0.01). Analyses were repeated twice with different passage numbers and had similar outcomes.

**Figure 5 cells-10-01278-f005:**
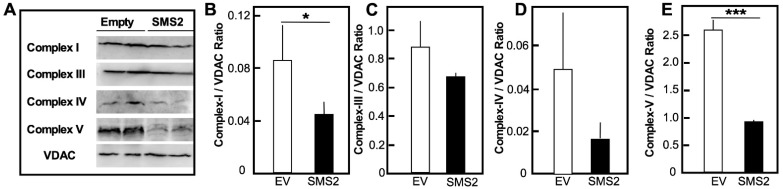
Levels of proteins of the electron transport chain. (**A**) Representative Western blot of cell homogenates of HepG2-EV (Empty) and HepG2-SMS2 (SMS2) cells. Images of full-sized gels are included in the Appendix A. (**B**–**E**). Quantification of band intensity normalized for the levels of VDAC. Average values ± SD (error bars) are shown (*n* = 3 dishes/point, * *p* < 0.05; *** *p* < 0.005). Data are representative of two independent experiments.

**Figure 6 cells-10-01278-f006:**
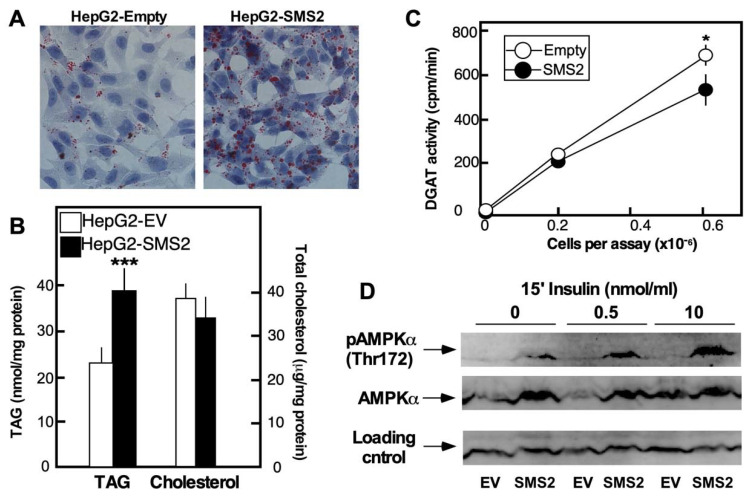
SMS2 overexpression is associated with excessive lipid droplet accumulation. HepG2-SMS2 and HepG2-EV cells were cultured under standard conditions in media containing 10% BSA and treated with insulin where indicated. (**A**) Formation of lipid droplets visualized by Oil Red-O staining. (**B**) Levels of TAG and total cholesterol. (**C**) DGAT activity measured as described under “Experimental Procedures”. (**D**) Western blot analyses of AMK phosphorylation pattern. Total AMPK and β-actin were used as independent loading controls. Mean values ± SD (error bars) are shown (*n* = 3 dishes/point, * *p* < 0.05; *** *p* < 0.01). Data are representative of four (**A**), one (**C**), and two (**B**,**D**) independent experiments. Images of full-sized blots are available in the Appendix A.

## Data Availability

The data presented in this study are available in the present manuscript and in the Appendix A. The results from replicative experiments are available upon request by the corresponding author.

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
