# Peer review of "Onset of Senescence and Steatosis in Hepatocytes as a Consequence of a Shift in the Diacylglycerol/Ceramide Balance at the Plasma Membrane"

_cells, 2021, doi:10.3390/cells10061278_

Round 1
Reviewer 1 Report
Major points:
It is difficult to understand how many independent experiments were done and how the data were statistically analyzed: the test(s) used (t-test?) should be stated in the Methods or the Figure legends; and the meaning of the asterisks should be explsained in the Figure legend. In Fig. 3B and D and Fig. 4C-E: it should be indicated in the Figure legends error bars indicate SD. And the number of independent experiments should be given.
Moreover: Apparently, the results shown in Fig. 1 are from a single experiments done in triplicate (representative for at least two independent experiments). In that case, the error bars represent only the accuracy of the measurement, but not the variability obtained with several independent experiments.
Similarly, according to legend to Figures 2,5 and 6: mean and SD always show results from n=3 dishes/point: does that mean a single experiment, done in triplicate?; or are these 3 independent experiments? Only in the latter case is it possible to conclude whether there are significant differences (or not) between SMS2 overexpressing cells and controls.
In general, several independent experiments, analyzed together using appropriate statistical tests are needed to draw any conclusions. If these independent experiments have been performed then the data form all experiments should be shown; or the authors should justify why it is not possible or necessary to do so.
The authors say that there was a trend for increased SM level in the SMS2 cells that did not reach statistical significance(line 216/217; Figure 2). Though this description is used in more and more publications when a p-value is slightly above 0.05, this reviewer believes this term is not appropriate. However, if the actual p-value is given, this reviewer does not see a major problem with it. The problem is that based on the mean values and error bars in Figure 2A to C, it appears unlikely that the p-value for a t-test would be close to 0.05 (with n = 3).
Additional points:
There are only few references in the introduction. Though some information may be well known for many readers it would be helpful to add more references (e.g. in the second paragraph of the introduction).
As stated in the Materials and Method section a single clone with the highest SMS2 expression was chosen for all experiments. Performing at least some critical experiments with an independent clone would help to exclude that some results obtained with the SMS2 clone are "clonal artifacts". If there are data obtained with independent clones they should also be presented; if not, the authors should at least discuss this point or argue why a "clonal artifact" is unlikely.
The Rhodamine-WGA of HepG2-SMS2 image does not fit to the overlay (also the Nomarski image appears to be shifted slightly relative to the overlay)
Minor points:
Delete first sentence in introduction (line 33/34).
Sequences of primers used for PCR cloning of SMS2 should be included in the Methods section so that readers are able to deduce the sequence of the final plasmid construct. The same applies to the SMS1-plasmid (or give reference for this plasmid).
Dilution and order number of the antibodies used should be given.
Legend to Fig.2 and Fig.3: the "beta"(greek symbol) for beta-actin and beta-galactosidase, respectively, is missing. In legend to Fig. 6 "b-actin" should be replaced by "beta(greek symbol)-actin".
Reviewer 2 Report
This investigation was aimed to determine the effect of sphingomyelin synthase (SMS) overexpression on the balance of diacylglycerol(DAG)/Ceramide at the plasma membrane in vitro. Authors show that SMS) overexpression increases DAG, downregulates PKC, and induces senescence in HepG2 cell line. The experimental strategy was well-executed and the evidences are quite clear. However some issues still need to be addressed.
In lines 390 – 391 authors state that complex I level was unaffected: “In contrast, the abundance of complex I and III was seemingly unaffected”; however, while WB analysis shows (Figure 5A) what authors state, the plot (Figure 5C) shows a significant difference. This issue must be carefully corrected and the sentence must be rewritten. Moreover, subsection B of Figure 5 is missing.
I believe that the following sentence at the beginning of the introduction section should not be included as part of it: “The introduction should briefly place the study in a broad context and highlight why it is important”.
The second paragraph of the introduction section does not include any source where the information was taken.
The transfection efficiency of the SMS vector is not indicated in materials and methods section.
The symbol “β“ for β-actin was not included in figure legend of figure 2; line 235.
Figure 3A in the following sentence is wrongly cited (line 222): “The only metabolite of the SMS reaction for which the differences reached statistical significance, was DAG (Fig. 3A), which increased by almost 500 pmol/mg. Pr. or by 50%”. Figure 3A stands for cell proliferation analysis.
Define “PC” at the first appearance.
Measurement unit in figure 3B (“y” axis) is not shown.
The symbol “β“ for β-galactosidase was not included in figure legend of figure 3; line 305.
Based on HUGO Gene Nomenclature Committee, gene and protein symbols must be indicated according to accepted nomenclatures. So, throughout the manuscript (main text, figures and figures legends) gene and protein symbols must be properly indicated. As reference, authors should review HUGO website as well as the following article: PMID: 22836666.
The following sentence is incomplete: “SASP is group of proteins that are secreted by senescence cells and facilitate the onset of inflammation in the tissues of the aging organisms [19] and/or exacerbate”. Line 318.
The following sentence is wrongly edited since appears along with figure legend 4: “DGAT, which acylates DAG to TAG and is a rate-limiting step in TAG synthesis was done…”. Line 367.
The following sentence is wrongly edited since appears along with figure legend 5: “The elevated AMPK phosphorylation is in line with the known functions of the kinase as an energy sensor activated by a drop in ATP levels, as seen in HepG2-SMS2 cells…”. Line 399.
The following sentence is wrongly edited since appears along with figure legend 5: “Finally, we also investigated the status of AMP kinase (AMPK), a focal point of regulation…”. Line 450.
Author Response
Please see the attached word document

Reviewer 3 Report
Onset of Senescence and Steatosis in Hepatocytes as a Consequence of a Shift in the Diacylglycerol/Ceramide Balance at the Plasma Membrane
The authors investigated modification of the bioactive lipids ceramide and DAG in HepG2 cells by transfecting them to overexpress SMS2. The hypothesis was overexpression of SMS2 increased the rate of senescence by way of mitochondrial respiration rate suppression. This manuscript used novel techniques coupled with established techniques to demonstrate SMS2 played a pivotal role in DAG homeostasis leading to cellular senescence via the disruption of the plasma membrane.
Comments:
- The authors could improve the understanding of the manuscript with improved grammatical precision. For example, the introduction seemed to be written by a person that is lacks the skill to create a story that the reader can follow. Writing a more succinct introduction will draw the interest of the reader and uncover the direction the authors are leading. Additionally, there are minor grammatical issue to address. For instance, the authors used a number of abbreviations that were not described. The authors could benefit from describing the abbreviations that will be carried out throughout the manuscript, such as SM, MEF, EMT, and PC. The reader will be required to continue the review to the methods to understand the difference with the abbreviations.
- The authors indicated the cells were grown to sub-confluency. That leaves to speculate as to whether the cells were grown anywhere from 20%-99% confluency. The number of cells will drastically change the outcome of the experiment, hence the authors should consider giving a percent confluency to assist in reproducibility.
- The experiments were carried out after transfection utilizing G418 at 2 mg/ml. There are reports indicating the necessity of reducing the concentration from 2 mg/ml to 1 mg/ml. The authors should consider informing the reader as to why they did not lower the concentration of the
- {Proc Natl Acad Sci U S A. 2020 Jul 21;117(29):17019-17030}
- {Mol Cells. 2019 Dec 31;42(12):869-883.}
- {Cancer Cell Int. 2020; 20: 139}
- Figure 3A used flow cytometry analysis to determine the cellular generation from parent to generation 1-7 being identified by 8 different shades of color. The authors will benefit from making the color choices more distinct. Once all the colors are on the same graph it is difficult to see which generation is being expressed. For example generation 3 and generation 7 seem to be either the same or generation is nonexistent. Additionally, when using red and yellow, when the overlap it become difficult to see which generation is being expressed. Taking into account a person that is color blind, these colors will not be visible for the individual reviewing the data.
- Figure 3C has arrow heads in one image but they are not present in the neighboring image. Data will be enhanced by elaborating on the purpose of the arrows and why they are absent in the neighboring image. Further, a scale marker should be considered to size of the image being viewed.
- The Western blots (Fig 5 and 6) has been cropped to eliminate the ability of the reader to see antibody specificity. Where possible, a full bot is suggested; otherwise, the authors are encouraged to show all bands surrounding the band of interest to indicate the specificity as well as how cuts on the blot was determined.
Author Response
Please see the attached word document

Reviewer 4 Report
This manuscript by Deevska et al entitle “Onset of senescence and steatosis in hepatocytes as a consequence of a shift in the diacylglycerol/ceramide balance at the plasma membrane” (Manuscript Number: cells-1185038) examined the impact of DAG and ceramide on cellular metabolism and senescence. Based on the scope of the journal, the topic of the research is of high interest and the study is valuable. Although this is an interesting work, there are several limitations in the present manuscript that should be improved.
Major concerns:
- Introduction should be improved. Authors focus mainly on the description of DAG, ceramide and SMS isoforms. It is a very descriptive introduction without delving into the innovative aspects of the manuscript. Authors should cover here the main topic of the article, the role of SMS in senescence.
- Materials and methods section is incomplete. There are several techniques that are not described, even mentioned in this section (western blot analysis, RNA extraction and quantitative gene expression analysis, cellular oxygen consumption or seahorse analysis…). Additionally, statistical analyses are not described.
- Authors mentioned in the results section that key genes responsible for SASP were measured. These results are not described here (lines 316 – 318). Moreover, data obtained from IL-1ß and NLPR3 is not a definitive evidence. Protein expression levels should be analyzed to further support the senescence-associate secretory phenotype.
- Authors state that “oxygen consumption rates (Fig. 4B-E) showed reduction in the mitochondrial basal and maximal respiration rates, the coupling efficiency and ATP production (lines 384-386)”. Are these changes statistically significant? I do not see asterisks on the graphs.
- Discussion should be improved. In this section mitochondrial metabolism, one of the most relevant finding of this article, is not well discussed. Regarding this, reviewer is interested to know if authors have measured the levels of sphingomyelin (SM) and cardiolipin on mitochondria. Simoes et al reported significant decreased levels of mitochondrial SM together with increased mitochondrial cardiolipin levels on steatotic liver from NAFLD mouse model, which was related to a defective oxidative phosphorylation (doi:10.3390/antiox9100995). This mitochondrial structural composition affects mitochondrial respiration that finally come to age (doi:10.1038/s41419-017-0105-5). These novel aspects of mitochondrial remodeling should be discussed in the text.
Minor concerns:
- Abbreviations should be corrected. Authors should explain each of their abbreviations the first it appears in the main text. For instance: DAG (line:34), PKC (line:40), PP2A (line:41) … are not explained. Or SM2 is defined in the line 67, instead of line 63. This problem persists throughout the entire manuscript. Only few abbreviations are well described.
- Authors must provide the reference number of all the materials used (reagents, kits, antibodies…). This information is missing.
- Line 128: SMS1-pcDNA or SMS2 -pcDNA?
- Line 222: Fig. 3A or Fig. 2D?
Round 2
Reviewer 1 Report
The authors addressed all important points. I have no further comments.
Reviewer 4 Report
I considered that the manuscript has improved considerably according to the suggestions. The article is novelty and contains potentially interesting data. Therefore, I recommend the publication in the present form.